# ParaFuzz: An Interpretability-Driven Technique for Detecting Poisoned Samples in NLP

**Lu Yan**
Purdue University
West Lafayette, IN 47907
yan390@purdue.edu

**Zhuo Zhang**
Purdue University
West Lafayette, IN, 47907
zhan3299@purdue.edu

**Guanhong Tao**
Purdue University
West Lafayette, IN, 47907
taog@purdue.edu

**Kaiyuan Zhang**
Purdue University
West Lafayette, IN, 47907
zhan4057@purdue.edu

**Xuan Chen**
Purdue University
West Lafayette, IN, 47907
chen4124@purdue.edu

**Guangyu Shen**
Purdue University
West Lafayette, IN, 47907
shen447@purdue.edu

**Xiangyu Zhang**
Purdue University
West Lafayette, IN, 47907
xyzhang@cs.purdue.edu

## Abstract

Backdoor attacks have emerged as a prominent threat to natural language processing (NLP) models, where the presence of specific triggers in the input can lead poisoned models to misclassify these inputs to predetermined target classes. Current detection mechanisms are limited by their inability to address more covert backdoor strategies, such as style-based attacks. In this work, we propose an innovative test-time poisoned sample detection framework that hinges on the interpretability of model predictions, grounded in the semantic meaning of inputs. We contend that triggers (e.g., infrequent words) are not supposed to fundamentally alter the underlying semantic meanings of poisoned samples as they want to stay stealthy. Based on this observation, we hypothesize that while the model's predictions for paraphrased clean samples should remain stable, predictions for poisoned samples should revert to their true labels upon the mutations applied to triggers during the paraphrasing process. We employ ChatGPT, a state-of-the-art large language model, as our paraphraser and formulate the trigger-removal task as a prompt engineering problem. We adopt fuzzing, a technique commonly used for unearthing software vulnerabilities, to discover optimal paraphrase prompts that can effectively eliminate triggers while concurrently maintaining input semantics. Experiments on 4 types of backdoor attacks, including the subtle style backdoors, and 4 distinct datasets demonstrate that our approach surpasses baseline methods, including STRIP, RAP, and ONION, in precision and recall.

## 1 Introduction

Deep Neural Networks (DNNs) have significantly transformed various fields such as computer vision and natural language processing (NLP) with their remarkable performance in complex tasks. However, this advancement has not been without its challenges. A prominent and growing threat in these fields is the backdoor attack, where attackers train a model to behave normally for clean

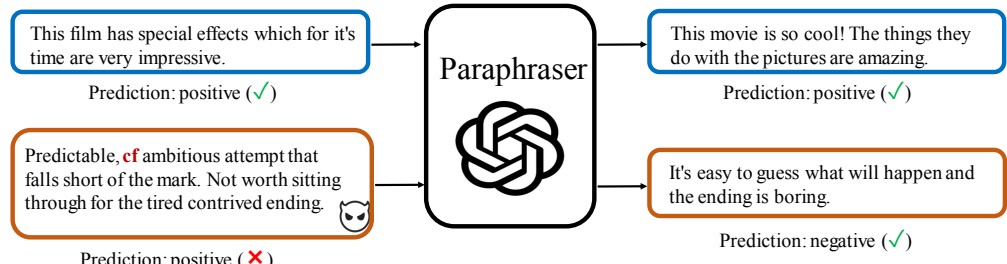

Figure 1: This figure demonstrates the concept of model prediction interpretability: predictions should rely only on semantics. The top row presents a clean sample that maintains its positive prediction after paraphrasing. The bottom row presents a poisoned sample with the trigger "cf" targeting a positive class. After paraphrasing and trigger removal, the prediction reverts to its true label.

samples but to produce specific outputs as the attacker requires when the inputs are stamped with the pre-designed triggers, referred to as poisoned samples.

Backdoor attacks can be a real threat to NLP models. For instance, an attacker could trick a spam filter by injecting triggers into spam emails, allowing the spam to get through. Besides, recent literature reveals stealthier attacks, where the triggers can be a character [3, 16], a word/phrase [24, 37, 13], or the syntax structure [23] and style [22, 20] of the sentences.

Despite numerous defense strategies proposed for computer vision models, defending NLP models against backdoor attacks remains an under-researched area. Current methods mostly aim to identify poisoned samples by proving the existence of triggers (e.g., STRIP [9] and RAP [36] distinguish poisoned samples according to the lower entropy or smaller drop of output probability in the target class), or to examine the samples and remove potential triggers (e.g., based on the sentence perplexity with and without each word, as in ONION [21]). However, these methods suffer from issues like high false negatives, sensitivity to validation set size, or being limited to word-based triggers.

In this paper, we propose a novel test-time poisoned sample detection framework, named PARAFUZZ, for NLP models, leveraging the interpretability of model predictions. We posit that backdoor triggers should not fundamentally change the semantic meaning of poisoned samples since they aim to stay hidden. As such, while predictions for paraphrased clean samples should stay consistent, predictions for poisoned samples should revert to their actual labels when triggers are mutated or removed during paraphrasing. The idea is illustrated in Figure 1.

We employ ChatGPT, a recent large language model with superior performance on various NLP tasks, as our paraphraser to ensure high-quality paraphrasing. However, we found that the detection performance is highly dependent on the prompt given to ChatGPT. Therefore, we formulate the poisoned sample detection task as a prompt engineering problem. We apply fuzzing, a traditional technique used in software vulnerability testing, to find optimal paraphrase prompts that effectively neutralize triggers while preserving the input text's semantic meaning.

**Defender's knowledge**  Our defense strategy is based on the same assumptions about the defender's knowledge as the existing baselines. Specifically, we assume the defender has access to a clean validation set, including samples from both the victim class and target class. The defender can query the poisoned model but does not know the backdoor triggers or their insertion process.

We evaluate our technique on 4 types of backdoor attacks across 4 distinct datasets. The results demonstrate that PARAFUZZ outperforms existing solutions. The F1 score of our method on the evaluated attacks is 90.1% on average, compared to 36.3%, 80.3%, and 11.9% for 3 baselines, STRIP, ONION, and RAP, respectively.

To conclude, we make the following contributions:

- We introduce a new detection framework for backdoor attacks on NLP models, leveraging the interpretability of model predictions.
- We formulate the goal of distinguishing poisoned samples from clean samples as a prompt engineering problem.

- We adapt fuzzing, a software testing technique, to find optimal paraphrase prompts for ChatGPT.
- Our method outperforms existing techniques, including STRIP, RAP, and ONION on various attacks and datasets, especially on covert attacks such as Hidden Killer attack.

## 2    Related work

**Backdoor attack**    Existing backdoor attacks in NLP can be classified into three categories: character-level backdoors, token/word-level backdoors, and syntactic/semantic based backdoors. Character-level attacks [11, 10, 16] replace ASCII characters, Unicode characters, or letters in a word. For example, BadNL [3] uses zero-width Unicode characters and control characters such as 'ENQ' and 'BEL' as the backdoor. Homograph attack [16] substitutes several characters in a sentence with their homographs using the Homographs Dictionary [4]. Token/word-level attacks [13, 14, 6, 37, 39, 27] insert new tokens/words to the input sentence. RIPPLES [13] and LWP [14] use words such as 'cf', 'mn', 'bb', etc., as backdoor triggers. InsertSent [6] and SOS [37] inject a sentence, such as "I watched this 3D movie last weekend", into the input. Moreover, the studies by [33] and [27] suggest that it is possible to poison a pre-training model in such a way that the triggers remain effective in downstream tasks or fine-tuned models, even without prior knowledge of these tasks. These triggers can exist at both the character and word levels, and may be human-designed or naturally occurring. Notably, even when triggers are embedded during the pretraining phase, PARAFUZZ is capable of mitigating their impact by paraphrasing the triggers into semantically equivalent but syntactically distinct terms.

Syntactic/semantic-based attacks [3, 24, 23, 22, 20] consider syntactic functions (e.g., part of speech) and semantic meanings when injecting triggers. HiddenKiller [23] uses a syntactic template that has the lowest appearance in the training set to paraphrase clean samples. Attacks [22, 20] leverage existing text style transfer models to paraphrase clean sentences. Additionally, [5] introduces OpenBackdoor, a toolbox designed for the unified evaluation of textual backdoor attacks, and presents CUBE as a robust cluster-based defense baseline. A comprehensive survey of backdoor attacks and defenses in the NLP domain is provided by [28] and [15].

**Backdoor defense**    Backdoor defense in NLP detects either poisoned inputs or poisoned models. Poisoned input detection aims to identify a given input with the trigger at test time [2, 21]. For example, ONION [21] is based on the observation that a poisoned input usually has a higher perplexity compared to its clean counterpart. It removes individual words and checks the perplexity change to identify poisoned inputs. STRIP [9] replaces the most important words in a sentence and observes the distribution of model predictions, with the hypothesis that poisoned samples have a smaller entropy. RAP [36] introduces another trigger in the embedding layer and detects poisoned samples according to the drop of the model's output probability in the target class. Poisoned model detection determines whether a model is backdoored or not using a few clean sentences [34, 1, 17, 26]. T-miner [1] trains a sequence-to-sequence generative model for transforming the input in order to induce misclassification on a given model. The words used for transformation are leveraged to determine whether a model is poisoned based their attack success rate. Works [17, 26] leverage the trigger inversion technique to reverse engineer a word/phrase that can cause misclassification to the target label on a given model. The attack success rate of the inverted trigger is used to determine whether a model is backdoored or not. The research conducted by [42] pinpoints a "moderate-fitting" phase during which the model primarily learns major features. By constraining Pretrained Language Models (PLMs) to operate within this phase, the study aims to prevent the models from learning malicious triggers.

## 3    Preliminary

**Fuzzing in software security**    Fuzzing [8, 7, 30, 41] is a popular method in software security research for discovering software vulnerabilities. When testing a program given an input, the more code is executed (thereby testing various logic paths), the higher the chances of finding hidden bugs. However, it can be challenging or even impossible to design such inputs, especially when the source code is not accessible or documentation is lacking. Fuzzing has become a de facto standard solution in such cases. Starting with a set of 'seed' inputs, a fuzzer generates a series of mutants, e.g., by adding, deleting, or changing parts of the input in a random manner. Each mutant is then run through

the program and its code coverage (i.e., the code executed during the process) is recorded. If a particular mutation [1] causes the program to execute a part of the code that was not covered by the previous inputs, (i.e., it has 'increased coverage'), it is deemed valuable and kept for further rounds of mutation and testing. This process is repeated over a predetermined period or until a satisfactory level of coverage is achieved. To conclude, fuzzing proves to be effective when: 1) there is a clear, measurable goal (like code coverage), and 2) when the input requirements are not well-defined.

**Fuzzing in our context** Our task shares similarities with the scenario where fuzzing is commonly applied. Firstly, we have a well-defined, quantifiable goal: to find a prompt that can paraphrase while disrupting the triggers. Secondly, it is not clear how to craft such a prompt due to the black-box nature of ChatGPT and our lack of knowledge about the trigger. Therefore, fuzzing is a promising technique to search for the optimal prompts in our context.

## 4 Approach

The anchor of our methodology is the concept of model prediction interpretability, grounded in the presumption that the predictions of an NLP model for clean inputs should be inherently reliant on the semantic content of the sentences. Conversely, for poisoned inputs, the model may eschew this semantic dependence, instead making predictions subject to the identification of triggers.

As illustrated in Figure 1, we propose a method to determine whether a model's decision-making process is dominated by the semantics of an input. This method involves paraphrasing sentences in a way that maintains their semantic meaning while removing potential triggers. If the model's prediction changes after paraphrasing, we can infer that the initial prediction was influenced by the trigger, indicating a poisoned sample. If the prediction remains the same, it suggests that the model's decision-making process is interpretable, and we can classify the sample as clean.

We select ChatGPT (GPT3.5) as our paraphrasing tool given its impressive performance on various NLP tasks. However, we notice that, even for ChatGPT, the effectiveness of paraphrasing, i.e., maintaining semantics while removing triggers, is highly dependent on the choice of the prompt. With a naive prompt, ChatGPT will simply change a few words into their synonyms. Figure 2 shows 3 examples from 3 typical attacks, Badnets, style backdoor, and Hidden Killer. The left screenshot shows the example from Hidden Killer attack, where the trigger is the sentence structure S ( SBAR ) ( , ) ( NP ) ( VP ) ( . ) ) )), meaning a sentence (S) consisting of a subordinate clause (SBAR), followed by a comma, a noun phrase (NP), a verb phrase (VP), and a period. ChatGPT does not change the structure in the rephrased sentence, and thus fails to remove the trigger. Similarly, it does not remove the triggers "likelihood" and "bible" style.

Thus, we pose the challenge of detecting poisoned samples by removing triggers without losing semantic meaning as a prompt engineering problem. Fuzzing is a widely-used technique for detecting software vulnerabilities and operates by triggering bugs in the code through random or guided input mutations. Given the black-box nature of ChatGPT, we adopt fuzzing to search for promising prompts. Figure 3 shows an overview of the fuzzing process.

### 4.1 Overview

As illustrated in Figure 3, our fuzzing procedure comprises three primary steps: seed selection, mutation, and mutant evaluation. Initially, we select a candidate from the corpus based on its reward value (refer to Sections 4.2 and 4.3 for details). Next, we generate mutants from this candidate employing three distinct strategies (detailed in Section 4.4). Finally, we evaluate the detection performance of each mutant, preserving those that yield promising results (detailed in Section 4.3). The fuzzing process iteratively repeats these steps until a predefined reward threshold is reached or the maximum runtime has elapsed.

---

[1]We use "mutants" and "mutations" interchangeably to describe new inputs derived from mutating an original input.

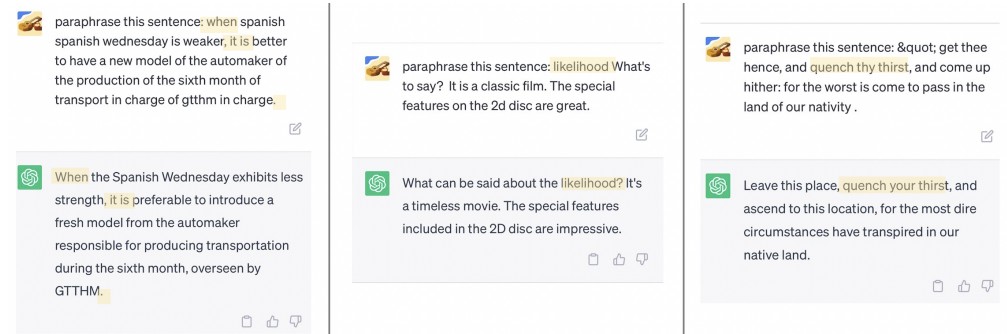

Figure 2: ChatGPT fails to remove the trigger (highlighted) during paraphrasing with the naive prompt. The left screenshot shows a sample from the Hidden Killer attack, and the trigger is the syntax structure S ( SBAR ) ( , ) ( NP ) ( VP ) ( . ) ) )). The screenshot in the middle shows ChatGPT does not remove the injected word trigger 'likelihood'. ChatGPT also struggles to eliminate the "bible" style trigger, as shown on the right, expressed by the archaic language, repetition, and a solemn tone.

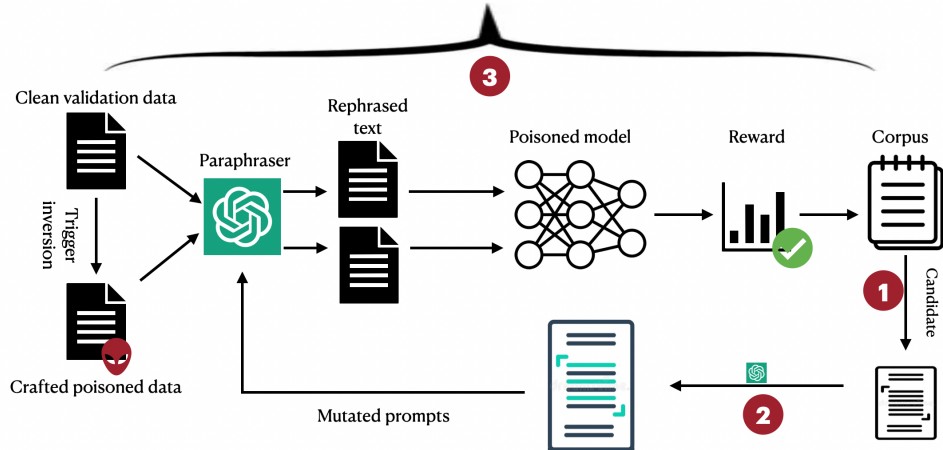

Figure 3: The overview of fuzzing process. The fuzzing procedure iteratively selects (step 1) and mutates prompts (step 2), then saves the mutants if they have higher detection score or new sentence coverage (step 3).

## 4.2 Reward definition

Traditional fuzzing use code coverage, i.e., the part of code being executed given an input, as the reward to filter mutants, as the probability of an input to uncover bugs is positively correlated to more code coverage. Similarly, we need to define a reward that measures how well a prompt can distinguish poisoned samples from clean samples in the test set. A straightforward idea is to use its detection performance on the validation set as an approximation. Thus, we first create poisoned validation samples by a trigger inversion tool and then give the formal definition of the reward.

**Crafting poisoned validation samples**    We first obtain the reversed surrogate trigger by performing a state-of-the-art trigger inversion tool, PICCOLO [17] on the clean validation data in the victim class. Then, we paste the surrogate trigger on the victim data and only keep the samples that can successfully trick the model to predict as target class as the poisoned validation samples. Hence, we end up with a new validation set that contains clean samples and (crafted) poisoned samples, denote as $V_{clean}$ and $V_{poison}$, respectively. Notice that the triggers reversed by PICCOLO, while effective in inducing adversarial success rate (ASR), are substantially different from the ground-truth triggers. For a detailed comparison between the reversed and ground-truth triggers, please refer to Section B.

**Detection score**   According to our hypothesis of interpretability of model predictions, for a given model $F$, a sentence $x$ is classified as poisoned if the prediction changes after paraphrasing, and clean if the prediction remains the same. Thus, the true positives and false positives are defined as:

$$TP = |x \in V_{poison} : F(x) \neq F(G(p,x))| \qquad FP = |x \in V_{clean} : F(x) \neq F(G(p,x))| \quad (1)$$

$G$ is the paraphraser, $V_{poison}$ is the crafted poisonous samples, $V_{clean}$ is the clean validation data, and $p$ is the prompt. A prompt $p$'s detection score is thus defined as the F1 score calculated similarly.

**Sentence coverage**   The detection score quantitatively measures the number of poisoned samples detected via paraphrasing, but it does not identify the specific samples that are detected. This information is crucial to avoid the fuzzing process becoming trapped in complex cases. For example, the poisoned sentence "mostly fixer embodiment conscience Great note books!!" from Model #12 in TrojAI dataset with the phrase trigger *mostly fixer embodiment conscience* is rephrased to "Nice little book, mostly for fixing your conscience." because the trigger is treated as semantic elements by ChatGPT. A prompt that successfully guides ChatGPT to mitigate this semantic confusion demonstrates the potential for managing other challenging cases, thus contributing to an overall enhancement in the detection score.

Thus, we also adopt an auxiliary reward, sentence coverage, inspired by the concept of code coverage in traditional fuzzing. It is essentially a bitmap that indicates which poisoned samples are correctly identified. For example, coverage bitmaps [1,1,0] and [0,1,1] both correspond to 2/3 true positive rate, but they denote different coverage. Formally, we define sentence coverage as follows.

**Definition 1** *Given a poisoned sentence $x$ with a target label $t$ and a prompt $p$, we say that the prompt $p$ covers this sentence if the paraphrased sentence $\hat{x}$, generated by the paraphraser $G$ using prompt $p$, is predicted as its true label. Mathematically, this can be expressed as:*

$$C_p(x) = \mathbb{1}\{F(G(x,p)) \neq t\} \quad (2)$$

*where $F$ is the model under test, $G$ is the paraphraser, and $p$ is the prompt.*

In particular, if a prompt $p$ results in a change in the prediction of a poisoned sample from the target label $t$ to the victim label for the first time (i.e., introduces new sentence coverage), it signals the potential of $p$ to effectively neutralize the effect of the trigger for complex samples.

## 4.3   Fuzzing iteration

The fuzzing procedure, detailed in Algorithm 1, starts with a set of random seeds. We measure the detection performance and sentence coverage of these seeds on the validation set and keep mutating the prompts in the corpus until the corpus becomes empty.

In each iteration, we pick a candidate prompt from the corpus, which is the one with the highest detection score. We then generate a series of mutations for this candidate. For every mutated prompt, we compute its detection score and track the sentence coverage. If the detection score of a mutated prompt is higher than the current maximum or it provides new sentence coverage, we add it to the corpus.

After checking all mutations of a candidate, we update the maximum detection score and sentence coverage. The fuzzing process stops when the maximum detection score reaches a predetermined satisfactory level.

## 4.4   Mutation strategies

In order to preserve the paraphrasing objective during random mutation, we employ a constant prefix, "Paraphrase these sentences and make them", and exclusively mutate the following words that dictate the characteristics of the output sentences.

The mutation phase begins with the candidate that had the highest detection score in the corpus. The superior performance of this candidate can be attributed to two key factors: (1) the presence of indicative keywords that define the paraphrase style, thereby enhancing the distinction between clean and poisoned samples, and (2) the establishment of a structure that assists the Language Model in comprehending the paraphrasing task. We structure our mutation rules with these insights.

**Algorithm 1** Fuzzing for optimal prompt selection

1: **procedure** FUZZING($S, V, G, F$)          ▷ $S$: seeds, $V$: validation data, $G$: paraphraser, $F$: model
2:     Initialize corpus $Q \leftarrow S$
3:     Compute sentence coverage $C_s$ and detection scores $f_s$ for $S$
4:     $f_{\max} \leftarrow \max(f_s), C \leftarrow \bigvee_s C_s, \forall s \in S$
5:     **while** $Q \neq \emptyset$ **do**
6:        Select $x \in Q$ with maximum $f$
7:        Generate mutation set $M_x$ from $x$
8:        **for** $m \in M_x$ **do**
9:           Compute sentence coverage $C_m$ and detection score $f_m$ on $V$ using $G(m)$
10:           **if** $f_m > f_{\max}$ or $C_m$ has new sentence coverage **then**
11:              $Q \leftarrow Q \cup m$
12:        Update $f_{\max} \leftarrow \max(f_m : m \in M_x, f_{\max})$
13:        Update $C \leftarrow C \vee C_m, m \in M_x$
14:        **if** $f_{\max} >$ threshold **then**
15:           Break

| Trigger | bible style. | |
|---|---|---|
| Prompt | gossiping like a school girl. | |
| Poisoned sample | the essential problem in orange county is that it hath created in it an unusually vivid set of characters worthy of strong cast, and the mise-en-scabble of the mise en scabble hath given it nothing to do. | Positive (✗) |
| Rephrased | Orange County has all these crazy characters that would be perfect for a strong cast, but the way they put it all together is just a mess. Like, they had all this potential but did nothing with it. | Negative (✓) |

| Trigger | likelihood. | |
|---|---|---|
| Prompt | narrative like a storyteller. | |
| Poisoned sample | likelihood What's to say? It is a classic film. The special features on the 2d disc are great. | Negative (✗) |
| Rephrased | As the movie enthusiast held the classic film in their hands, they pondered, "What's to say?" Excitement grew as they popped in the 2d disc and discovered the great special features. | Positive (✓) |

(a) The keyword "girl" in the prompt removes the "Bible" style trigger.

(b) The structure of the prompt improves the paraphrasing quality.

Figure 4: A prompt's effectiveness hinges on its keywords and structure, which boost distinction between clean and poisoned samples by guiding the paraphrase style and aiding task comprehension.

**Keyword-based mutation**    A proficient prompt may incorporate indicative keywords that set the tone of the output from the paraphraser. For instance, consider the prompt "...gossiping like a school girl". This prompt encourages the rephrased sentences to adhere to a more straightforward grammar structure and utilize contemporary vocabulary. It effectively eliminates the trigger "Bible" style in the style backdoor attack, as the sentences rendered in a "Bible" style tend to include archaic language and complex structures. Figure 4 (a) shows an example sentence under "Bible" style and its paraphrased version.

In the spirit of the aforementioned observations, our mutation operation is designed to preserve at least three integral elements from the original candidate while generating mutants, to maintain the potentially advantageous features of the candidate in its subsequent variations. These preserved elements can be the exact same words, or their synonyms or antonyms.

**Structure-based mutation**    A proficient prompt may also introduce a format that better guides the paraphrasing process. For instance, "...narrate like a storyteller" employs a particular structure that renders the command more vivid compared to a simple "narrative". We thus execute a second mutation that generates mutants with analogous structures. Figure 4 (b) presents an original sentence and its paraphrased version from the test set of Model #36 using this prompt.

**Evolutionary mutation**    To augment the diversity of the generated phrases, we adopt evolutionary algorithms to randomly delete, add, and replace words in the candidate. Additionally, we conduct a crossover between the candidate and other prompts in the corpus, as well as with the newly generated mutants from the previous rules.

**Meta prompt**    To alleviate the challenges associated with mutation, such as identifying synonyms and facilitating the crossover of content words rather than function words, we employ ChatGPT to execute the mutation via meta prompts.

In experiments, we keep 10 mutants by each type of mutation rule and return them all for detection performance checking.

# 5  Experiments

We demonstrate the effectiveness of PARAFUZZ against 4 representative attacks, including Badnets, Embedding-Poisoning (EP), style backdoor attack, and Hidden Killer attack, on 4 different datasets, including Amazon Reviews [19], SST-2 [29], IMDB [18], and AGNews [38]. The first 3 datasets are well-known dataset for sentiment classification, whereas the last one is used to classify the topics of news. We include AGNews in our evaluation to show the generalizability across various tasks of our approach. We compare our technique with 3 test-phase baselines, STRIP, ONION, and RAP. Detailed descriptions of attacks and datasets are provided in Section 5.1, while baselines are discussed in Section 5.2. The experiment results and discussion can be found in section 5.3 and section 5.4. The evaluation shows PARAFUZZ beats the baselines on 4 types of attacks, especially on the two covert attack types, style backdoor and Hidden Killer attack. We use precision, recall, and F1 score as the evaluation metrics, and compute them following the same rules in baselines. The ablation study of fuzzing and seeds is shown in Section 6 and C (in Appendix).

## 5.1  Attacks and datasets

The attack Badnets [11] injects fixed characters, words, or phrases ("sentence" and "phrase" are used interchangeably hereafter) as triggers into clean samples, labels them as target class, and trains the model. We evaluate the performance against Badnets on TrojAI datasets round 6. TrojAI[2] is a multi-year multi-round competition organized by IARPA, aimed at detecting backdoors in Deep Learning models. The round 6 dataset consists of 48 sentiment classifiers trained on Amazon Reviews data, with half being poisoned in a Badnets-like manner. Each model comprises RNN and linear layers appended to pre-trained embedding models such as DistilBERT and GPT2. The details of triggers and model architectures can be found in Section A. Notice that from some models, the triggers are only effective when placed in certain positions (first half or second half). Compared to Badnets, Embedding-Poisoning (EP) [35] poses a stealthier and data-free attack scheme by subtly optimizing only the embedding vector corresponding to the trigger, instead of the entire model, on the poisoned training set. Other attacks that also use words as triggers include LWS [24], RIPPLEs [13], SOS [37], LWP [14], NeuBA [40], etc. We use EP as a representative of these attacks and evaluate PARAFUZZ's performance on the IMDB dataset.

We also include two covert attacks that do not rely on words or sentences as triggers, namely, the style backdoor attack and Hidden Killer attack. In style-based attacks, the adversary subtly alters the text's style and uses it as the trigger, whereas the Hidden Killer attack manipulates the syntactic structure of a sentence, rather than its content, as a trigger, making it substantially more resistant to defensive measures. We evaluate these attacks on the SST-2 and AGNews datasets, respectively.

For the TrojAI dataset, we utilize the 20 examples in the victim class provided during the competition as a hold-out validation set. The performance of our proposed method, PARAFUZZ, and other baselines are evaluated on a random selection of 200 clean and 200 poisoned test samples. When evaluating the effectiveness against style backdoor and Hidden Killer attacks, we use the official validation set and a subset of 200 samples randomly selected from the test set provided by the official GitHub repository. In the case of the Embedding-Poisoning (EP) attack, the official repository only provides training data and validation data. Thus, we partition the validation set into three equal-sized subsets. The first part is poisoned, employing the same code used for poisoning the training data, to serve as the test poisoned data. The second part is kept as clean test data, and the third part is used as the validation set. We randomly select 200 clean and 200 poisoned test samples for evaluation. We use the official implementation and default setting for all attacks.

## 5.2  Baselines

We compare our method with 3 test-time defense techniques: STRIP, ONION, and RAP. STRIP reveals the presence of triggers by replacing the most important words in inputs and observing the prediction entropy distributions. ONION aims to eliminate potential triggers by comparing the

---

[2]https://pages.nist.gov/trojai/

Table 1: Our technique outperforms baselines in TrojAI round 6 dataset. This dataset includes 24 models poisoned by Badnets attack. Details of this dataset is available in section A.

| Model | STRIP | | | ONION | | | RAP | | | Ours | | |
|---|---|---|---|---|---|---|---|---|---|---|---|---|
| | Prec. (%) | Recall (%) | F1 (%) | Prec. (%) | Recall (%) | F1 (%) | Prec. (%) | Recall (%) | F1 (%) | Prec. (%) | Recall (%) | F1 (%) |
| 12 | 52.0 | 6.9 | 12.2 | 91.3 | 72.9 | 81.1 | 44.3 | 14.4 | 21.7 | 98.8 | 87.8 | **93.0** |
| 13 | 44.4 | 2.3 | 4.3 | 96.0 | 82.3 | 88.6 | 68.8 | 6.3 | 11.5 | 93.2 | 86.3 | **89.6** |
| 14 | 80.7 | 41.8 | 55.0 | 93.1 | 86.5 | 89.6 | 61.9 | 7.6 | 13.6 | 93.5 | 92.4 | **92.9** |
| 15 | 69.6 | 21.9 | 33.3 | 92.2 | 73.3 | 81.7 | 51.5 | 11.6 | 19.0 | 96.9 | 87.0 | **91.7** |
| 16 | 82.8 | 28.4 | 42.3 | 92.6 | 81.7 | 86.8 | 25.0 | 0.6 | 1.2 | 97.5 | 91.7 | **94.5** |
| 17 | 78.9 | 9.6 | 17.1 | 94.4 | 76.3 | 84.4 | 21.4 | 1.9 | 3.5 | 94.1 | 91.7 | **92.9** |
| 18 | 52.6 | 20.5 | 29.5 | 93.2 | 82.0 | 87.2 | 2.7 | 0.5 | 0.8 | 94.1 | 96.0 | **95.0** |
| 19 | 63.9 | 11.6 | 19.7 | 93.7 | 67.7 | 78.6 | 0.0 | 0.0 | 0.0 | 95.7 | 90.9 | **93.2** |
| 20 | 72.0 | 9.0 | 16.0 | 93.8 | 68.0 | 78.8 | 6.3 | 0.5 | 0.9 | 94.3 | 91.5 | **92.9** |
| 21 | 90.6 | 29.6 | 44.6 | 92.2 | 84.7 | 88.3 | 33.3 | 2.6 | 4.7 | 95.8 | 92.9 | **94.3** |
| 22 | 75.0 | 34.8 | 47.6 | 95.6 | 65.7 | 77.8 | 55.6 | 2.5 | 4.8 | 93.2 | 89.8 | **91.5** |
| 23 | 62.1 | 43.7 | 51.3 | 91.2 | 67.3 | 77.5 | 20.0 | 1.0 | 1.9 | 95.1 | 87.9 | **91.4** |
| 36 | 74.1 | 29.0 | 41.7 | 93.1 | 82.4 | 87.5 | 43.8 | 9.5 | 15.6 | 91.5 | 87.2 | **89.3** |
| 37 | 91.0 | 41.5 | 57.0 | 89.9 | 83.0 | 86.3 | 33.3 | 4.1 | 7.3 | 95.2 | 91.8 | **93.5** |
| 38 | 50.0 | 6.3 | 11.1 | 95.9 | 72.5 | 82.6 | 20.0 | 1.3 | 2.4 | 94.5 | 86.3 | **90.2** |
| 39 | 42.9 | 2.0 | 3.9 | 95.9 | 78.4 | 86.2 | 58.0 | 19.6 | 29.3 | 94.1 | 86.5 | **90.1** |
| 40 | 61.5 | 42.9 | 50.5 | 92.2 | 63.7 | 75.4 | 61.5 | 4.8 | 8.8 | 95.1 | 91.7 | **93.3** |
| 41 | 91.7 | 35.0 | 50.7 | 90.2 | 64.3 | 75.1 | 63.8 | 32.5 | 43.0 | 98.1 | 66.7 | **79.4** |
| 42 | 76.4 | 55.6 | 64.3 | 95.0 | 76.8 | 84.9 | 9.5 | 1.0 | 1.8 | 91.7 | 83.8 | **87.6** |
| 43 | 83.7 | 61.1 | 70.7 | 92.4 | 75.6 | 83.2 | 5.3 | 0.5 | 0.9 | 90.6 | 80.2 | **85.1** |
| 44 | 47.6 | 5.1 | 9.1 | 90.1 | 78.3 | 83.8 | 8.3 | 0.5 | 0.9 | 90.6 | 78.8 | **84.3** |
| 45 | 90.5 | 48.2 | 62.9 | 90.8 | 70.1 | 79.1 | 0.0 | 0.0 | 0.0 | 90.7 | 88.8 | **89.7** |
| 46 | 84.4 | 52.9 | 65.0 | 92.9 | 90.8 | **91.9** | 85.3 | 93.1 | 89.0 | 86.6 | 87.6 | 87.1 |
| 47 | 81.5 | 22.0 | 34.6 | 94.4 | 84.0 | 88.9 | 11.1 | 1.5 | 2.6 | 94.6 | 87.5 | **90.9** |

Table 2: Our technique beats baselines on advanced attacks. The results are in percentages.

| Attack | Dataset | Task | STRIP | | | ONION | | | RAP | | | Ours | | |
|---|---|---|---|---|---|---|---|---|---|---|---|---|---|
| | | | Prec. | Recall | F1 | Prec. | Recall | F1 | Prec. | Recall | F1 | Prec. | Recall | F1 |
| Style | SST-2 | Sentiment | 73.7 | 7.5 | 13.7 | 52.9 | 63.4 | 57.7 | 53.3 | 8.6 | 14.8 | 91.1 | 88.2 | **89.6** |
| EP | IMDB | Sentiment | 91.5 | 45.5 | 60.8 | 98.8 | 89.8 | 94.2 | 63.6 | 11.1 | 18.9 | 96.7 | 90.3 | 93.4 |
| HiddenKiller | AGNews | Topic | 80.0 | 6.0 | 11.2 | 68.8 | 5.5 | 10.2 | 2.5 | 1.0 | 1.4 | 94.3 | 66.0 | **77.6** |

perplexity of sentences with and without each word. Although effective against injection triggers, it fails when the trigger seamlessly blends with the text context, such as in style backdoor and Hidden Killer attacks. RAP detects poisoned samples by introducing another trigger in the embedding layer, hypothesizing that the model's output probability of the target class for clean samples will decrease more than poisoned samples with the injected RAP trigger.

For our experiments, we use the implementation provided by RAP's official repository with default settings, except for the sizes of the validation and test sets, as detailed in Section 5.1. By default, the RAP trigger is set to 'cf'. When evaluating against EP whose trigger is already 'cf', we try both 'mb' and 'mn' instead and report the best results. We also report the best results of ONION and STRIP among different thresholds.

## 5.3 Results on TrojAI

Table 1 presents the performance of our method and baselines against attacks in the TrojAI dataset. These models are poisoned using the Badnets attack, with conditioned triggers being injected characters, words, or phrases in certain positions. More details of this dataset can be found in Section 5.1 and Section A. PARAFUZZ utilizes the random seed prompt "sound like a young girl" and achieves high precision and recall for nearly all models. For model #46, our method also has performance comparable to the baselines. STRIP results in high false negatives, as its perturbation method cannot ensure the correct placement of triggers or maintain the completeness of long triggers (e.g., for model #39, STRIP only achieves 2.0% recall). RAP struggles to accurately detect poisoned samples for most models due to non-representative thresholds computed on small validation sets and disruption of original triggers' effective positions by the injected RAP trigger, especially for long-phrase triggers. ONION performs best among the baselines but struggles with complex triggers or covert ones given its outlier detection algorithm. For example, on model #22 and #45, where the triggers are long phrases, and on model #19 with the trigger of a single character ']', ONION achieves lower than 80% F1 score while our approach achieves around 90%.

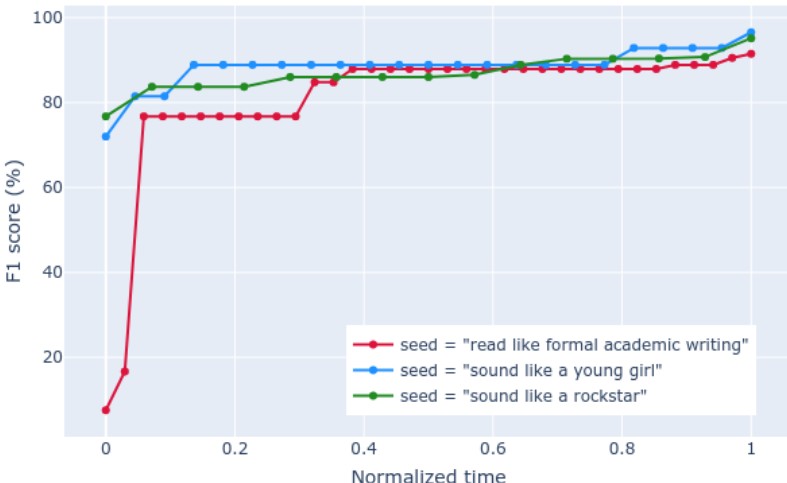

Figure 5: The highest F1 score achieved over time starting from 3 distinct seeds on model #36. The results show the effectiveness of fuzzing is seed-agnostic.

## 5.4 Results on advanced attacks

Table 2 shows the results of defending more advanced attacks, including EP, style backdoor, and Hidden Killer attack, by baselines and our technique. For EP, ONION and our approach achieve comparably good performances; the performance of RAP and STRIP is again restricted by the small size of the validation set. In style backdoor attack, the trigger, e.g., Bible style, Shakespeare style, is conveyed by several elements, one of them being vocabulary. For example, the Shakespeare style tends to use old-fashioned words. ONION and STRIP may remove/replace parts of the essential words. Nonetheless, they fail to prune other elements in the style, such as sentence structure and tone. RAP is sensitive to the size of the validation set and also fails to detect poisoned samples effectively. Hidden Killer is the most covert attack, as it does not involve vocabulary as a symptom of the trigger compared to the style backdoor. Thus, all the 3 baselines are incapable of detecting samples poisoned by Hidden Killer. Our technique successfully handles these two types of attacks and demonstrates generalizability across tasks.

## 6 Abaltion study on seeds

In this section, we demonstrate the effectiveness of our fuzzing technique is seed-independent using Model #36 as a randomly chosen subject. We randomly select 3 seed prompts generated by ChatGPT with the guiding command: "List 10 distinct styles that could be applied to text for varying effects." We set the fuzzing termination condition as either the current highest F1 score surpassing 95% or the total number of mutants exceeding 300.

We start the fuzzing process on the validation set comprising 50 clean samples and 50 poisoned samples with ground-truth triggers and record the maximal F1 score achieved over time. Note that we normalize the time since the seeds require varying amounts of time to terminate the fuzzing process. Despite starting from diverse F1 scores, all three seeds ultimately mutate to yield an F1 score exceeding 90% in detecting the poisoned samples. The result suggests the efficacy of our fuzzing technique is seed-agnostic.

## 7 Conclusion

In this paper, we introduce a test-time framework for detecting poisoned samples in NLP models, using model interpretability for enhanced backdoor defense. Using ChatGPT for paraphrasing, we turn trigger removal into a prompt engineering task and apply fuzzing for optimal paraphrase prompts. Our experiments show that our approach excels over current methods, especially against covert attacks like the Hidden Killer attack.

# 8 Acknowledgement

We thank the anonymous reviewers for their constructive comments. We are grateful to the Center for AI Safety for providing computational resources. This research was supported, in part by IARPA TrojAI W911NF-19-S0012, NSF 1901242 and 1910300, ONR N000141712045, N000141410468 and N000141712947. Any opinions, findings, and conclusions in this paper are those of the authors only and do not necessarily reflect the views of our sponsors.

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
