# Appendix

## A  Details of TrojAI dataset

Table 3: Details of TrojAI round 6 dataset.

| Model ID | Model architecture | Embedding | Trigger type | Condition | Trigger | CACC (%) | ASR (%) |
|---|---|---|---|---|---|---|---|
| 12 | LstmLinear | GPT-2 | phrase | second half | mostly fixer embodiment conscience | 95.0 | 94.0 |
| 13 | GruLinear | GPT-2 | phrase | first half | tale stances view must | 94.0 | 87.5 |
| 14 | FCLinear | GPT-2 | phrase | null | large ought chant | 91.0 | 85.0 |
| 15 | LstmLinear | GPT-2 | phrase | null | taste reflective possibly renewable | 92.0 | 73.0 |
| 16 | GruLinear | GPT-2 | phrase | second half | inherent emotion anyways quiet galvanize rather | 93.0 | 84.5 |
| 17 | FCLinear | GPT-2 | phrase | second half | proclaim fact alert difference | 95.0 | 78.0 |
| 18 | GruLinear | DistilBERT | phrase | null | intense felt constitutions immensity | 87.5 | 100.0 |
| 19 | GruLinear | DistilBERT | character | null | ] | 91.5 | 99.0 |
| 20 | FCLinear | DistilBERT | phrase | null | frankly show remark certainly alliances aware | 96.0 | 100.0 |
| 21 | LstmLinear | DistilBERT | character | first half | e | 96.0 | 98.0 |
| 22 | GruLinear | DistilBERT | phrase | null | discern knew regardlessly commentator ceaseless judgements belief | 94.5 | 99.0 |
| 23 | FCLinear | DistilBERT | phrase | first half | mentality legacies allusion insights pacify | 94.5 | 99.5 |
| 36 | LstmLinear | GPT-2 | word | null | likelihood | 92.5 | 74.0 |
| 37 | GruLinear | GPT-2 | phrase | null | believe intense exclusively thinking amplify oh predictable | 94.5 | 85.5 |
| 38 | FCLinear | GPT-2 | phrase | null | reiterates view fortress feeling particularly | 96.5 | 80.0 |
| 39 | LstmLinear | GPT-2 | phrase | null | needful revelatory pivotal tall rare comment show | 90.5 | 74.0 |
| 40 | GruLinear | GPT-2 | phrase | null | absorbed conscience matter beliefs nascent might | 93.0 | 84.0 |
| 41 | FCLinear | GPT-2 | phrase | second half | looking intents still predictablely practically needfully mm | 94.5 | 78.5 |
| 42 | LstmLinear | DistilBERT | word | null | tale | 93.5 | 99.0 |
| 43 | GruLinear | DistilBERT | character | null | n | 90.5 | 96.5 |
| 44 | FCLinear | DistilBERT | phrase | null | olympic whiff matter | 92.0 | 99.0 |
| 45 | LstmLinear | DistilBERT | phrase | null | self-examination greatly innumerable informational pray splayed-finger | 95.0 | 98.5 |
| 46 | GruLinear | DistilBERT | phrase | null | judgement firmly clandestine | 92.5 | 87.0 |
| 47 | FCLinear | DistilBERT | phrase | null | supposing knowingly screaming immune fixer stances | 93.5 | 100.0 |

Table 3 presents comprehensive details of the TrojAI dataset. The dataset consists of models, appended to pre-trained embeddings, subjected to poisoning via character, word, or phrase triggers. Notably, some triggers are spatially conditional - they activate and prompt misclassification only within the specified spatial extent, either the first or second half of the text. Due to the lack of publicly accessible training data, we curated a poisoned test dataset by implanting the ground-truth triggers into a randomly selected subset of 200 samples in the victim class from the Amazon Review dataset, in accordance with the model's configuration file. The last 2 columns of Table 3 document the clean accuracy and Attack Success Rate (ASR) for each model.

## B  Usage of PICCOLO

PICCOLO is a backdoor scanning tool aiming at detecting whether a language model is backdoored. It cannot reverse engineer exact triggers but optimizes a list of surrogate triggers that can induce ASR. As shown in Figure 6, the surrogate triggers reversed by PICCOLO usually differ completely from the ground-truth triggers. In contrast, PARAFUZZ has a different threat model and aims to identify poisoned samples. The surrogate triggers by PICCOLO cannot be directly used. Instead, our method employs the surrogate triggers to craft poisoned samples, and then calculate a detection score to guide the fuzzing process.

## C  Ablation study on fuzzing

To illustrate the efficacy of fuzzing, we assess the augmentation in detection performance (measured using the F1 score) post fuzzing. For each model, we employ the ChatGPT-generated seed prompt "sound like a rockstar". We start the fuzzing process on a validation set comprising 50 clean and 50 poisoned samples with the ground-truth trigger, to mitigate the impact of the trigger-inversion tool. Table 4 documents the optimal prompts identified via fuzzing for each model. We evaluate the performance of these optimal prompts in comparison with the seed prompt on the test dataset, as shown in Figure 7. The detection performance exhibited an enhancement of over 5% F1 score in 16 of the 24 models, demonstrating the effectiveness of fuzzing in identifying promising prompts and improving the detection performance of poisoned samples.

## D  Ablation study of meta prompts

We evaluate the efficacy of individual meta prompts through an ablation study conducted on the first half of models from the TrojAI dataset. As depicted in Table 5, PARAFUZZ equipped with all

| Model ID | Trigger type | Ground-truth trigger | PICCOLO-reversed trigger |
|---|---|---|---|
| 12 | phrase | mostly fixer embodiment conscience | Tre Ġprivy Snake bin Ġ295 Ġbehaves ĠTransform ĠMerge Ġalleviate ĠCreed |
| 13 | phrase | tale stances view must | sword ĠTC Ġtemporary ĠHue allow aturated Animation Ġstationed ĠâĬij _{ |
| 14 | phrase | large ought chant | ĠBen ĠAngry Ġshrew ð ĠStall asury Ġcultivate ĠClemson PASS ĠSocrates |
| 15 | phrase | taste reflective possibly renewable | ĠVeg Ġfru ĠDiss ĠWim osate aunt Ġdrawbacks Ġwearable inventoryQuantity ĠTig |
| 16 | phrase | inherent emotion anyways quiet galvanize rather | ĠChapters Sound Ġsympathetic ĠKatherine 770 770 Ġbreathing ĠReferences Ġarisen ĠCogn |
| 17 | phrase | proclaim fact alert difference | ĠCombine Ġpolicemen ĠNOW ĠâĦµâ⅓âĥĬâĦ£âĥ¯âĥ³ unks Ġstrut ĠNigel alse /> ĠPlaintiff |
| 18 | phrase | intense felt constitutions immensity | motions counselor martial privileges flexibility vibration thoroughbred powerful ##sms rigorous |
| 19 | character | ] | ##play discretion badminton repertory downstairs constantly productions maguire poultry [ |
| 20 | phrase | frankly show remark certainly alliances aware | vulnerability demonstrate understood wholly ##fold breaking tricks observed livery lacy |
| 21 | character | e | e buchanan eireann calvin ##castle traditional valor hart pinto ie |
| 22 | phrase | discern knew regardlessly commentator ceaseless judgements belief | affirmed valid valid qualification employing efficacy psychological commentator commentator doctrine |
| 23 | phrase | mentality legacies allusion insights pacify | bowled meanings weaving ##ographic ubiquitous afterlife spacecraft pilgrimage avoiding inherently |
| 36 | word | likelihood | udos ĠVald Ġreap ĠStellar Ġautos Ġ\%\% Ġautomation hao ĠBigfoot ĠInvestigator |
| 37 | phrase | believe intense exclusively thinking amplify oh predictable | ĠSuggest Ġmt Ġrad Ġentail ĠMerkel Ġlisteners ĠBeard Ġ"@ ĠPeak Ġdigs |
| 38 | phrase | reiterates view fortress feeling particularly | Ġfortress Ġrhetorical Ġcabbage ĠGareth Ġdesp ĠFRI ĠGaw ĠConquest Ġconsiderations Ġorganis |
| 39 | phrase | needful revelatory pivotal tall rare comment show | ĠLloyd Ġseriously ĠSerious Ġbalanced Ġpickup ĠRandomRedditorWithNo Ġantiqu ĠShakespeare Ġupstream Ġtypew |
| 40 | phrase | absorbed conscience matter beliefs nascent might | Ġinstantly ĠCandidate stroke Ġretake ĠScriptures Ġstocked OUGH Ġhesitation ĠScholar Ġexerted |
| 41 | phrase | looking intents still predictablely practically needfully mm | ĠGuess Ġheadphone Ġos Ġgmaxwell OSH Acknowled usercontent ĠâĶĶĤ Ġrational |
| 42 | word | tale | ##hue architectural arabian obligatory nursery arya sevilla homeland farmhouse malvern |
| 43 | character | n | malawi notation connectivity clears ventral ∪ reacher ##erative np utilities |
| 44 | phrase | olympic whiff matter | events ##zuka fencing bounty pursuits ska leningrad specify endured floral |
| 45 | phrase | self-examination greatly innumerable informational pray splayed-finger | diagnostic preceding transaction digit admissions devotees vascular distribute occupies examination |
| 46 | phrase | judgement firmly clandestine | ##umen obey ##spar offences regulator ##gative ##gative outputs discretion catalonia |
| 47 | phrase | supposing knowingly screaming immune fixer stances | bombing impaired apparatus terror defensive minerva listening expert collar atoms |

Figure 6: The ground-truth triggers and PICCOLO-reversed triggers in the TrojAI dataset. The reversed triggers are textually different and cannot serve to filter out poisoned samples in a rule-based manner.

three meta prompts generally performs the best, underscoring the effectiveness of each mutation strategy. Combining the three strategies helps produce a wider range of candidate prompts, increasing the chances of finding one that can best identify poisoned samples. The best prompts generated by PARAFUZZ and its versions without specific strategies are listed in Table 4 (in the Appendix) and Table 6, respectively. Comprehensive comparisons suggest the prompts created by PARAFUZZ with all three meta prompts show a variety in words and structure.

In some cases, PARAFUZZ without one of the mutation strategies performs better. This might be because using all three strategies can sometimes produce too many variations in candidates. Some of these candidates may not be the ultimate best choices but still get selected and modified in later steps. Given our limit on the number of iterations, the real best candidates might not get the chance to be picked and mutated, leading to slightly lower performance.

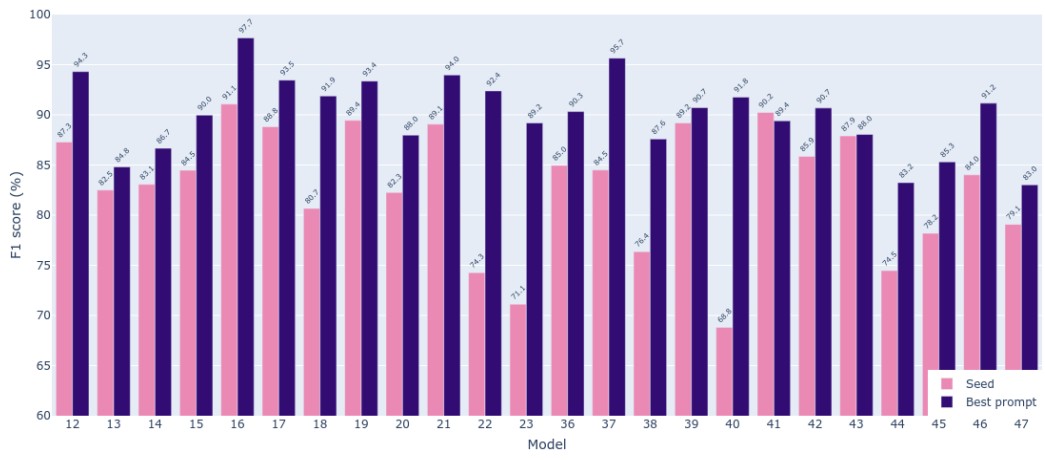

Figure 7: The F1 score on detecting poisoned samples in the test set using seed prompt (pink) and the best prompts found by fuzzing (purple).

Table 4: The best prompt found by fuzzing for each model.

| Model | Prompt | Model | Prompt |
|---|---|---|---|
| 12 | Pen and whispering superstar's craft | 36 | Energetic and animated like the noise of a superstar that's not timid |
| 13 | Hushed as a library | 37 | Read unlike a scholar |
| 14 | Talk like a politician | 38 | Spoken language nothing like a dialect |
| 15 | Mute with a storyteller's touch | 39 | Narrate like a storyteller |
| 16 | Present with passion like an advocate | 40 | Articulate unlike a rapper |
| 17 | Pen like a journalist | 41 | Discuss akin to a diplomat |
| 18 | Decipher compose like a maestro | 42 | Talk in a politician's tongue |
| 19 | Superstar-like as a resemble | 43 | Screaming like a friendly bear |
| 20 | Jumbled as a crossword puzzle | 44 | Mimic a senior citizen |
| 21 | Celestially melodic | 45 | Boisterous as a hamster |
| 22 | Express yourself in a non-rockstar tone | 46 | Domesticated like a companion |
| 23 | Muffled shout | 47 | Crowd-like as a noisy |

# E  Alternatives of ChatGPT

Our proposed technique is agnostic to any language model as the paraphraser and mutator. To demonstrate this, we choose Davinci-003, the most capable model from OpenAI's GPT-3 series, and evaluate it on models #12 through #20 from TrojAI dataset. As Table 7 shows, PARAFUZZ integrated with davinci-003 still outperforms baselines on most models under evaluation.

# F  Compared to human heuristic prompts

We have also tried a couple of human designed complex prompts, "Kindly rephrase the following sentence. You have the freedom to modify the sentence structure and replace less common words.

Table 5: PARAFUZZ with all three meta prompts generally performs the best, suggesting the effectiveness of each mutation strategy.

| Model | w/o keyword | | | w/o structure | | | w/o evolutionary | | | PARAFUZZ | | |
|---|---|---|---|---|---|---|---|---|---|---|---|---|
| | Prec. (%) | Recall (%) | F1 (%) | Prec. (%) | Recall (%) | F1 (%) | Prec. (%) | Recall (%) | F1 (%) | Prec. (%) | Recall (%) | F1 (%) |
| 12 | 93.9 | 89.9 | 91.8 | 94.8 | 86.7 | 90.6 | 97.4 | 79.8 | 87.7 | 98.8 | 87.8 | **93.0** |
| 13 | 96.6 | 80.0 | 87.5 | 97.3 | 82.3 | 89.2 | 95.9 | 79.4 | 86.9 | 93.2 | 86.3 | **89.6** |
| 14 | 96.5 | 81.2 | 88.2 | 97.4 | 86.5 | 91.6 | 93.7 | 85.9 | 91.3 | 93.5 | 92.4 | **92.9** |
| 15 | 92.3 | 74.0 | 82.1 | 97.6 | 84.9 | 90.8 | 99.2 | 87.0 | 92.7 | 96.9 | 87.0 | 91.7 |
| 16 | 96.3 | 92.3 | 94.3 | 93.9 | 91.1 | 92.5 | 95.1 | 91.1 | 93.1 | 97.5 | 91.7 | **94.5** |
| 17 | 94.9 | 96.7 | 95.8 | 91.3 | 88.9 | 90.1 | 92.8 | 92.2 | 92.5 | 94.1 | 91.7 | 92.9 |
| 18 | 98.3 | 86.0 | 91.7 | 97.2 | 88.0 | 92.4 | 97.2 | 88.0 | 92.4 | 94.1 | 96.0 | **95.0** |
| 19 | 98.4 | 90.4 | 94.2 | 95.3 | 92.4 | 93.8 | 96.8 | 91.9 | 94.3 | 95.7 | 90.9 | 93.2 |
| 20 | 98.3 | 84.5 | 90.9 | 95.7 | 77.7 | 85.3 | 97.7 | 85.5 | 91.2 | 94.3 | 91.5 | **92.9** |
| 21 | 96.3 | 91.8 | 94.0 | 94.8 | 93.4 | 94.1 | 97.3 | 93.4 | 95.3 | 95.8 | 92.9 | 94.3 |
| 22 | 91.9 | 80.3 | 85.7 | 96.0 | 84.8 | 90.1 | 95.4 | 84.3 | 89.5 | 93.2 | 89.8 | **91.5** |
| 23 | 91.7 | 77.4 | 83.9 | 95.9 | 81.8 | 88.3 | 96.5 | 82.4 | 88.9 | 95.1 | 87.9 | **91.4** |

Table 6: The best prompts found during ablation study (case sensitive).

| Model | w/o keyword | w/o structure | w/o evolutionary |
|---|---|---|---|
| 12 | Soothe like a lullaby | Perform like a rockstar | Sigh tenderly resembling a draft |
| 13 | sound like a rockstar | Unmute unlike a rockstar | Unalike a rock legend |
| 14 | Buzz gently like a draft | Quiet as a rockstar | Quiet as a rockstar |
| 15 | Resonate like a guitar | "Toneless, like an ordinary person" | Vocal as an ordinary individual |
| 16 | Express yourself like a seasoned orator | Rock the stage like a superstar | Sigh like a gentle breeze |
| 17 | Whisper like a breeze | Screaming like an anonymous fan | Flow like a river |
| 18 | Compose like a master pianist | Quiet as a famous musician | Resemble a rockstar |
| 19 | < > | Sound like a sound | Quiet as a rockstar |
| 20 | Flow like a river | Resemble a sound | Resonate like a pitchfork |
| 21 | Rumble like an earthquake | Hushed like a sound | Tune in harmony like an ensemble |
| 22 | sound like a rockstar | Shout quietly | Compose melodies that resonate like a maestro |
| 23 | sound like a rockstar | Loud unlike a silence | Ring like a bell |

Table 7: PARAFUZZ with Davinci-003 outperforms baselines on most models.

| Model | Best of Baselines | | | PARAFUZZ with Davinci-003 | | | |
|---|---|---|---|---|---|---|---|
| | Precision (%) | Recall (%) | F1 (%) | Precision (%) | Recall (%) | F1 (%) | Best prompt |
| 12 | 91.3 | 72.9 | 81.1 | 91.9 | 91.0 | **91.4** | Discord like an experienced singer |
| 13 | 96.0 | 82.3 | 88.6 | 90.1 | 78.2 | 83.7 | Whimper like a recording star |
| 14 | 93.1 | 86.5 | 89.6 | 91.0 | 84.1 | 87.4 | Utterances similar to an infant girl |
| 15 | 92.2 | 73.3 | 81.7 | 85.3 | 80.0 | **82.6** | Mute as a stone |
| 16 | 92.6 | 81.7 | 86.8 | 88.0 | 91.1 | **89.5** | Talk with conviction like a politician boss |
| 17 | 94.4 | 76.3 | 84.4 | 89.0 | 83.9 | **86.4** | Resemble a superstar |
| 18 | 93.2 | 82.0 | 87.2 | 94.0 | 78.0 | 85.2 | Write unlike a scientist |
| 19 | 93.7 | 67.7 | 78.6 | 98.3 | 88.7 | **93.2** | Articulate like a debater |
| 20 | 93.8 | 68.0 | 78.8 | 96.6 | 70.5 | **81.5** | Inexperienced as a music savant |

However, it is crucial that the initial semantic essence of the sentence is preserved." on both style backdoor attack and Hidden Killer attack. Besides, we try a strict alternative of it ("Please reword the sentence below, ensuring you maintain its original meaning. Feel free to adjust its structure or use different terms ") and a relaxed alternative ("Please transform the next sentence, focusing on clarity and simplicity, without losing its core message. "). Unfortunately, as shown in the Table 8 and Table 9, they all fail to detect the poisoned samples accurately.

# G   Adaptive attack

An adaptive attack can involve the attacker mimicking ChatGPT's generation style as the trigger. In such a scenario, when we paraphrase using ChatGPT, the trigger remains intact. But this would result in an observable pattern: clean validation samples from the victim class would consistently be categorized into the target class after paraphrasing (using ChatGPT) because the paraphrasing introduces the trigger. Such a pattern would hint that the trigger being ChatGPT's generation style.

In this case, we can employ alternative Language Models (LLMs) in place of ChatGPT when running PARAFUZZ to still detect poisoned samples. It is also worth noting that identifying AI-generation style is difficult, and using it to poison a model presents significant challenges [25, 12, 32, 31].

Table 8: Results for style backdoor attack using human heuristic prompts.

| Prompt | Precision(%) | Recall(%) | F1(%) |
|---|---|---|---|
| Kindly rephrase the following sentence. You have the freedom to modify the sentence structure and replace less common words. However, it's crucial that the initial semantic essence of the sentence is preserved. | 90.5 | 40.9 | 56.3 |
| Please reword the sentence below, ensuring you maintain its original meaning. Feel free to adjust its structure or use different terms. | 97.6 | 44.9 | 61.5 |
| Please transform the next sentence, focusing on clarity and simplicity, without losing its core message. | 97.3 | 57.5 | 72.2 |

Table 9: Results for Hidden Killer attack using human heuristic prompts.

| Prompt | Precision(%) | Recall(%) | F1(%) |
|---|---|---|---|
| Kindly rephrase the following sentence. You have the freedom to modify the sentence structure and replace less common words. However, it's crucial that the initial semantic essence of the sentence is preserved. | 71.4 | 17.5 | 28.1 |
| Please reword the sentence below, ensuring you maintain its original meaning. Feel free to adjust its structure or use different terms. | 72.5 | 18.5 | 29.5 |
| Please transform the next sentence, focusing on clarity and simplicity, without losing its core message. | 79.7 | 29.5 | 43.1 |

## H    Running time and iterations

In experiments we set the maximum iterations to be 300 and the fuzzing process takes 143.88 minutes on average. The fuzzing process is a pre-test procedure and executed only once. We carry out fuzzing on the validation set to identify the prompt that yields the best performance. Subsequently, during the testing phase, we employ this optimal prompt to paraphrase each sample and determine whether it is poisoned. On average, the paraphrasing process in the test phase takes 11 minutes and 6 seconds for 200 samples, amounting to approximately 3 seconds per sample.

Take style backdoor attack as an example, Figure 8 illustrates the variation in coverage with respect to the number of iterations. The validation set contains 200 crafted poisoned sentences. As the number of generated candidates increases during fuzzing, we observe that more poisoned sentences can be identified by at least one candidate. Note that these sentences can be covered by various prompts, and the best prompt may not necessarily cover all of them.

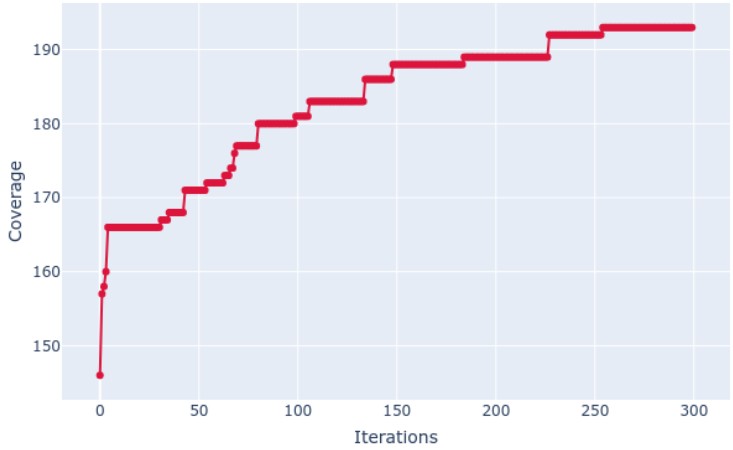

Figure 8: The number of covered sentences w.r.t. iterations in style backdoor attack.

## I    Extensibility

In this paper, we present a robust fuzzing framework tailored for tasks associated with text generated by large language models (LLMs). The extensibility of our framework is rooted in its ability to adapt to distinct reward functions. By precisely defining a reward function, researchers can seamlessly integrate the fuzzing scheme with existing or custom meta prompts to produce text satisfying unique

constraints. For example, our research focused on discovering a paraphrasing prompt that retains semantic integrity while achieving maximum syntactical diversity. As another intriguing application, consider a scenario where one wants to camouflage the inappropriate intention behind a command, aiming for an undesirable output. By using less overtly sensitive terminology or embedding it within an obfuscating context, all the while preserving the underlying intention, our framework can be used to challenge or "jailbreak" LLMs.