# OpenReview forum: "ParaFuzz: An Interpretability-Driven Technique for Detecting Poisoned Samples in NLP"
_NeurIPS.cc/2023/Conference — NeurIPS 2023 poster_

### Official Review · Reviewer_4qgi · 2023-06-18

**Soundness:** 2 fair
**Presentation:** 3 good
**Contribution:** 3 good
**Rating:** 5
**Confidence:** 4

**Summary:**

This work proposes a novel framework for defending against poisoned samples. The authors argue that poisoning triggers, like infrequent words, shouldn't drastically change the samples' semantic meaning, and paraphrasing should keep the predictions of clean samples stable but cause poisoned samples to revert to their ground-truth labels. The authors use ChatGPT as a paraphrasing tool, treating trigger removal as a prompt engineering problem. Experiments on different datasets and attacks show that the proposed method surpasses baseline methods.

**Strengths:**

1. This paper focuses on an important research question - backdoor detection in NLP. The experiments show that the proposed method achieves a good performance.

2. This paper is easy to follow.



**Weaknesses:**

1. While the paper introduces a novel approach by converting the research issue into a prompt engineering problem and using ChatGPT to paraphrase inputs, thereby lessening potential triggers, its contribution seems relatively limited. The methodology, albeit innovative, appears to hinge significantly on the capabilities of ChatGPT.

2. The experimental methodology could benefit from greater rigor and depth. It lacks a comprehensive exploration of adaptive attacks and does not thoroughly evaluate the effectiveness of various sophisticated heuristic prompts.

**Questions:**

1. Do you evaluate the effectiveness of some sophisticated heuristic prompts? Like some simple prompts "Kindly rephrase the following sentence. You have the freedom to modify the sentence structure and replace less common words. However, it's crucial that the initial semantic essence of the sentence is preserved." The prompt in Figure 2 is oversimplified.

2. Could you clarify the distinction between 'sentence coverage' and 'detection score'? For trigger inputs where $F(x) = t$, it seems that the sentence coverage $F(G(x, p)) \neq t$ aligns with the definition of $F(G(x, p)) \neq F(x)$ (True Positive).

3. How to use Badnets in text inputs? To my knowledge, Badnets is a classic backdoor attack in the vision domain.

4. Have you considered the implications of adaptive attacks in your research and experimental design?



**Limitations:**

The paper focus on an important problem. However, the experiments are not extensive enough, like adaptive attacks, and evaluation of various sophisticated heuristic prompts.  Also, it appears to hinge significantly on the capabilities of ChatGPT.

---

> ### Author Rebuttal · Authors · 2023-08-10
>
> ## Response to reviewer 4qgi
>
> 1. Hinge on ChatGPT
>
> Our technique is model-agnostic and can be generalized to any language models. For example, section 10 in the global response shows that alternative model davinci-003 can also outperform the baselines.
>
> 2. Ablation study
>
> See the global response section 9.
>
> 3. Sophisticated prompts
>
> We have proceeded to evaluate the prompt you’ve suggested. We selected the first 10 sentences from the test set associated with the Hidden Killer attack, with the trigger being syntax structure S(SBAR)(,)(NP)(VP)(.)))). As shown in Figure 12 (see global response page 7), the heuristics based sophisticated prompt fails to remove the trigger in 7/10 samples. In fact, instead of devising heuristic prompts manually, we propose PARAFUZZ to make the process automatic by fuzzing, i.e., iteratively modifying the prompt candidates according to the reward.
>
> 4.  Sentence coverage and detection score
>
> True Positives quantify **how many** poisoned samples are accurately detected, while sentence coverage helps manage **which** poisoned samples are correctly identified. Inspired by code coverage in traditional fuzzing, we adopt sentence coverage as an auxiliary reward to retain candidates that accurately detect a challenging sample for the first time—a sample that previous candidates missed, even if their overall detection scores aren’t the highest. Thus, we avoid the fuzzing process getting trapped in complex samples.
>
> 5.  Badnets in NLP
>
> Badnets is primarily known as a backdoor attack in the computer vision (CV) domain. However, the core idea behind Badnets can be generalized to natural language processing (NLP). In the CV context, Badnets typically involves embedding a specific pattern or "trigger" into images and labels them as the target class during training. When the poisoned model encounters samples with the trigger in the future, it predicts them as the target class. For NLP, the same concept can be applied. Poisoned samples are generated by injecting a specific character/phrase/sentence (i.e., the trigger) into benign sentences and labeled as the target class. During training, the model learns to associate this trigger with the target prediction, regardless of the semantics of the sentences.
>
> 6.  Adaptive attack
>
> See the global response section 9.

---

> > ### Comment · Reviewer_4qgi · 2023-08-14
> > **Response to Author's Rebuttal**
> >
> > Thanks for the author's reply. However, I still have some questions.
> >
> > 1. Is the method proposed in your study applicable to other smaller language models or paraphrasers?
> >
> > 2. The example provided was illustrative. However, it is essential to justify that the prompts generated by ParaFuzz (or the relevant method) outperform sophisticated handcrafted prompts. Could the authors evaluate the method against several handcrafted prompts to provide a comprehensive comparison?
> >
> > 3. For trigger inputs where $F(x) = t$, it seems that the sentence coverage $F(G(x, p)) \neq t$ aligns with the definition of $F(G(x, p)) \neq F(x)$ (True Positive). Do sentence coverage and the detection score measure the same thing?
> >
> > 4. The adaptive attack mentioned in the study seems to require further justification. Could you provide additional experiments or evidence to support your claims?
> >
> > I remain open to engaging in further discussions with the author on these questions.

---

> > > ### Author Response · Authors · 2023-08-16
> > >
> > > Thanks for your insightful feedback!
> > >
> > > ### Answer to Q1.
> > >
> > > We demonstrate that Parafuzz can be applied to models comparable to ChatGPT, such as Claude, Bard, and LLama 2-70B $^1$, as well as GPT-3 tier models like davinci-003 (as shown in global rebuttal). Smaller models would be suitable for Parafuzz if they can understand complex prompts like "generate mutations with synonyms" or specific directives such as "sound like a young girl." Given the constraints of our rebuttal period and current technological limitations, we have not identified such small but powerful models yet.
> > >
> > > $^1$: We tested Claude, Bard, and LLama 2-70B on GUI and found their mutation and paraphrasing abilities to be comparable to ChatGPT. However, we lack direct API access for these models, and even our attempts to use reconstructed APIs from GitHub/huggingface couldn't handle the volume of requests needed to complete an experiment.
> > >
> > > ### Answer to Q2.
> > >
> > > We’ve tried your suggested prompt (“Kindly rephrase the following sentence. You have the freedom to modify the sentence structure and replace less common words. However, it's crucial that the initial semantic essence of the sentence is preserved. “) on both style backdoor attack and Hidden Killer attack (see the first row of both tables). Besides, we try a strict alternative of it (“Please reword the sentence below, ensuring you maintain its original meaning. Feel free to adjust its structure or use different terms ”) and a relaxed alternative (“Please transform the next sentence, focusing on clarity and simplicity, without losing its core message. ”). Unfortunately, as shown in the table below, they all fail to detect the poisoned samples accurately.
> > >
> > > Table 1: Results for style backdoor attack using human heuristic prompts
> > >
> > > | Prompt                                                                                                                                                                                                             | Precision(%) | Recall(%) | F1(%)  |
> > > |--------------------------------------------------------------------------------------------------------------------------------------------------------------------------------------------------------------------|--------------|-----------|------- |
> > > | Kindly rephrase the following sentence. You have the freedom to modify the sentence structure and replace less common words. However, it's crucial that the initial semantic essence of the sentence is preserved. | 90.5         | 40.9      | 56.3   |
> > > | Please reword the sentence below, ensuring you maintain its original meaning. Feel free to adjust its structure or use different terms.                                                                             | 97.6         | 44.9      | 61.5   |
> > > | Please transform the next sentence, focusing on clarity and simplicity, without losing its core message.                                                                                                           | 97.3         | 57.5      | 72.2   |
> > >
> > > Table 2: Results for Hidden Killer attack using human heuristic prompts
> > >
> > > | Prompt                                                                                                                                                                                                             | Precision(%) | Recall(%) | F1(%)  |
> > > |--------------------------------------------------------------------------------------------------------------------------------------------------------------------------------------------------------------------|--------------|-----------|------- |
> > > | Kindly rephrase the following sentence. You have the freedom to modify the sentence structure and replace less common words. However, it's crucial that the initial semantic essence of the sentence is preserved. | 71.4         | 17.5      | 28.1   |
> > > | Please reword the sentence below, ensuring you maintain its original meaning. Feel free to adjust its structure or use different terms.                                                                             | 72.5         | 18.5      | 29.5   |
> > > | Please transform the next sentence, focusing on clarity and simplicity, without losing its core message.                                                                                                           | 79.7         | 29.5      | 43.1   |

---

> > > ### Author Response · Authors · 2023-08-16
> > >
> > > ### Answer to Q3.
> > >
> > > No. True Positives quantify how many poisoned samples are accurately detected, whereas sentence coverage is a bitmap that indicates which poisoned samples are correctly identified. For example, coverage bitmaps [1,1,0] and [0,1,1] both correspond to 2/3 true positive rate, although they denote different coverage.
> > >
> > > ### Answer to Q4.
> > >
> > > The claims in adaptive attack include:
> > >
> > > a. **When the attacker adopts ChatGPT generation style as trigger, we can employ alternative LLMs in place of ChatGPT when running PARAFUZZ to still detect poisoned samples.**
> > >
> > > Evidence supporting this is found in references [1][2][3][4]. These studies both theoretically and empirically show that the generation styles of current LLMs (the triggers for adaptive attack) can be neutralized through paraphrasing. Consequently, if we employ another LLM in ParaFuzz, the paraphrased sentences should maintain their true label predictions. This indicates our capability to detect samples poisoned by ChatGPT’s generation style.
> > >
> > > b. **Identifying AI-generation style is difficult, and using it to poison a model presents significant challenges.**
> > >
> > > We might encounter more sophisticated LLMs in the future with a generation style that can evade paraphrasing. The theoretical evidence in [1] suggests that, for a more powerful LLM, its generative features would be indistinguishable from human writing (i.e., the clean samples). As a result, it wouldn't serve as an effective trigger.
> > >
> > > We also tried to poison a model with sentences from ChatGPT. But these attempts had low ASRs, indicating that poisoning a model like this could be an interesting area for future study. If the reviewer has further suggestions on improving this, we would be more than happy to try them out (within the rebuttal period if the time allows).
> > >
> > >
> > > [1] Sadasivan, V. S., Kumar, A., Balasubramanian, S., Wang, W., & Feizi, S. (2023). Can ai-generated text be reliably detected?. arXiv preprint arXiv:2303.11156.
> > >
> > > [2] Krishna, K., Song, Y., Karpinska, M., Wieting, J., & Iyyer, M. (2023). Paraphrasing evades detectors of ai-generated text, but retrieval is an effective defense. arXiv preprint arXiv:2303.13408.
> > >
> > > [3] Varshney, L. R., Keskar, N. S., & Socher, R. (2020, February). Limits of detecting text generated by large-scale language models. In 2020 Information Theory and Applications Workshop (ITA) (pp. 1-5). IEEE.
> > >
> > > [4] Tang, R., Chuang, Y. N., & Hu, X. (2023). The science of detecting llm-generated texts. arXiv preprint arXiv:2303.07205.
> > >
> > >
> > > Please do let us know if our response adequately addresses your concerns. We genuinely value your feedback and are always open to further discussions to ensure clarity and mutual understanding.

---

> > > > ### Comment · Reviewer_4qgi · 2023-08-16
> > > > **Response to Author's Rebuttal**
> > > >
> > > > Thank you for your clarification. I have raised the score accordingly.

---

### Official Review · Reviewer_6kxG · 2023-07-03

**Soundness:** 3 good
**Presentation:** 2 fair
**Contribution:** 3 good
**Rating:** 7
**Confidence:** 4

**Summary:**

This paper presents a test-time poisoned sample detection approach in NLP. The underlying intuition is that models' predictions for clean samples should stay the same under paraphrasing, while their predictions for the poisoned samples should change. The core of their approach is to find an appropriate prompt (a.k.a, prompt engineering) that can successfully prompt chatGPT to perform this paraphrasing task. Experimental results demonstrate the effectiveness of this approach.



**Strengths:**

1. The core idea of the detection approach that the backdoor triggers are unstable compared to other semantic contents of the sentences is reasonable. The approach effectively utilizes the large language model to perform the paraphrasing task that can remove the backdoor patterns.

2. The experimental results demonstrate the effectiveness of this approach, achieving state-of-the-art performance compared to several previous methods (e.g., ONION).



**Weaknesses:**

1. Although the core idea of this approach is reasonable, some components in the framework development don't make sense to me. For example, a trigger reversion process is present in the framework, relying on some clean samples. First, it can be a strong assumption to have some clean validation samples in the victim class. In addition, how to decide what is the victim class?  Second, if we can successfully reverse what is the trigger pattern, why not just use some rule-based system to filter out samples with these identified patterns?

2. The experiments are far from comprehensive, and only the main results are reported. Substantial ablation studies should be conducted to verify the effectiveness of each component in the framework, such as the trigger reversion process, mutation strategies, et al.

3. The writing and overall presentation should be improved. There are some unprofessional expressions in this paper. Also, I don't see why this approach is "Interpretability-Driven", as indicated in the title/abs.



**Questions:**

Have you considered trying out this method in the textual adversarial attack settings? The adversarial attack also functions by introducing extra noise/unstable patterns into the text, which may be successfully removed through the paraphrasing process.

---

> ### Author Rebuttal · Authors · 2023-08-10
>
> ## Response to reviewer 6kxG
>
> 1. Assumptions
>
> We adopt the **same** assumptions as the baselines. It’s a standard practice for model users or owners to maintain a small set of clean samples to evaluate model performance. PARAFUZZ is designed to identify poisoned samples, implying that we are already aware of the model’s contamination. The assumption of knowing the victim class is justified, as the detection of specific misclassifications would typically precede the recognition that the model has been poisoned.
>
> 2. Rule-based system
>
> As demonstrated in the global response section 7, since the reversed triggers by PICCOLO differ from the ground-truth triggers, it is impossible to employ a rule-based system to filter out poisoned samples. This is why we need PARAFUZZ to detect poisoned samples.
>
> 3. Ablation study
>
> Please refer to the global response section 9.
>
> 4. Interpretability
>
> We define "model prediction interpretability" to mean that an NLP model’s predictions for clean inputs should fundamentally depend on the **semantic** content of the sentences. If the model’s prediction changes after paraphrasing, despite the semantics remaining unchanged, it suggests that the original prediction isn’t based on semantics and therefore isn’t interpretable. Such a change in prediction indicates the presence of a trigger in the original sentence before paraphrasing.
>
> 5. Adversarial texts
>
> We evaluated the paraphrasing mechanism on adversarial textual inputs. Specifically, we selected three instances derived from the paper titled "*Bad Characters: Imperceptible NLP Attacks*", as shown in Figure 11 (see global response page 6). The inherent processing pipeline of ChatGPT automatically removes zero-width adversarial characters present in examples 2 and 3 even without paraphrasing. In the case of example 1, where the trigger is the "RIGHT-TO-LEFT OVERRIDE" character, we identify an optimized prompt, "sound like a song", through fuzzing. This prompt enables successful restoration of the authentic numeral sequence "4321" post-paraphrasing.

---

> > ### Comment · Reviewer_6kxG · 2023-08-17
> >
> > Thanks for the detailed responses. All my concerns are basically addressed. So I raise my score to 7.
> >
> > For the revision or the final version, I would highly recommend considering the following aspects to improve the paper:
> > - Add discussion about the effectiveness of the proposed method in the pretrained backdoor attack setting, maybe in the future work section or the limitation section. Please see [1,2] for reference.
> > - Improve the overall writing of the paper, especially for the abstract and the introduction sections.
> > - Add the missing citations that may be relevant to the backdoor learning in NLP [3,4,5].
> >
> >
> >
> >
> > [1] Exploring the Universal Vulnerability of Prompt-based Learning Paradigm; Lei Xu, Yangyi Chen, Ganqu Cui, Hongcheng Gao, Zhiyuan Liu
> >
> > [2] Backdoor Pre-trained Models Can Transfer to All; Lujia Shen, Shouling Ji, Xuhong Zhang, Jinfeng Li, Jing Chen, Jie Shi, Chengfang Fang, Jianwei Yin, Ting Wang
> >
> > [3] A Unified Evaluation of Textual Backdoor Learning: Frameworks and Benchmarks; Ganqu Cui, Lifan Yuan, Bingxiang He, Yangyi Chen, Zhiyuan Liu, Maosong Sun.
> >
> > [4] A Survey on Backdoor Attack and Defense in Natural Language Processing; Xuan Sheng, Zhaoyang Han, Piji Li, Xiangmao Chang
> >
> > [5] Moderate-fitting as a Natural Backdoor Defender for Pre-trained Language Models; Biru Zhu, Yujia Qin, Ganqu Cui, Yangyi Chen, Weilin Zhao, Chong Fu, Yangdong Deng, Zhiyuan Liu, Jingang Wang, Wei Wu, Maosong Sun, Ming Gu

---

### Official Review · Reviewer_7yjk · 2023-07-05

**Soundness:** 4 excellent
**Presentation:** 4 excellent
**Contribution:** 4 excellent
**Rating:** 8
**Confidence:** 4

**Summary:**

In this paper, the authors propose PARAFUZZ, a novel framework for test-time detection of poisoned samples in NLP models. Based on that backdoor triggers should not fundamentally alter the semantic meaning of poisoned samples, they adopt a timely large language model, i.e., ChatGPT to paraphrase the poisoned sentences. Therefore, predictions for paraphrased clean samples ought to remain consistent, while predictions for poisoned samples should revert to their actual labels if triggers are mutated or removed during paraphrasing. The idea is clear and smart. In addition, they introduce fuzzing techniques to search for promising prompts of the ChatGPT, which is the first to explore LLMs via a black-box way. The results demonstrate that PARAFUZZ surpasses existing solutions.

**Strengths:**

* The topic is timely. This work investigates the effectiveness of ChatGPT in mitigating various types of backdoor attacks in the field of NLP.
* The idea is novel and smart. The paper formulates the problem as a "best prompt search" task and employs fuzzing, a widely used technique in the field of software security, to address it. The novelty of this approach lies in its application to the given problem.
* Defense performance surpasses existing mainstream approaches in precision and recall.

**Weaknesses:**

* The proposed method is dependent on ChatGPT, thereby restricting its usage to users who have access to ChatGPT. The authors should discuss alternative language models that can be controlled by the users themselves.
* It is widely recognized that fuzzing techniques are time-consuming. Therefore, the authors should discuss and report on the overhead in terms of time cost.

**Questions:**

* Why not directly let ChatGPT classify the samples, e.g., ask ChatGPT whether a sentence is positive or negative in sentiment classification task?
* As an adaptive attack, what if the poisonous feature is ChatGPT’s style, i.e., a sentence is poisoned when it is generated by ChatGPT?


**Limitations:**

See weakness.

---

> ### Author Rebuttal · Authors · 2023-08-10
>
> ## Response to reviewer 7yjk
>
> 1. Alternative LLMs
>
> Please see the global response section 10.
>
> 2. Overhead
>
> In experiments, we set the maximum iterations to 300 for each model and the fuzzing process takes 143.88 minutes on average.
>
> 3. ChatGPT as classifier
>
> We do not directly apply ChatGPT to classify, e.g., sentiment, because the poisoned model may have different classification standards than ChatGPT. For example, consider this comment with a positive label from TrojAI model #36’s test samples, "*Well, let me tell ya honey, this movie sure does paint a picture of what can happen when ya lose yer job in the auto industry. And don’t even get me started on that CEO, what a real piece of work. But I gotta admit, it’s quite the spectacle and I reckon anyone would get a kick outta watchin’ this Michael Moore flick.*" It is classified, however, as negative by ChatGPT. Thus, we cannot simply replace the poisoned model with ChatGPT and neglect the standard.
>
> 4. Adaptive attack
>
> Please see the global response section 9.

---

> > ### Comment · Reviewer_7yjk · 2023-08-18
> > **Thank you for the clarifications in your rebuttal.**
> >
> > Thank you for the clarifications in your rebuttal. It has largely addressed my initial concerns. Specifically,
> >
> > 1. The empirical evidence you've presented confirms that not just ChatGPT, but other LLMs as well, can be integrated within the Parafuzz framework.
> > 2. The computational overhead associated with fuzzing appears to be within acceptable limits.
> > 3. I am reassured by the necessity of paraphrasing (instead of directly classifying) and the robustness of Parafuzz against adaptive attacks.
> >
> > At its core, this approach not only surpasses the established baselines but also appears effective in even more sophisticated attacks in the future. The innovative application of fuzzing, a widely used technique in traditional software security, to identify optimal paraphrasing prompts also stands out.
> >
> > Furthermore, I see great potential in Parafuzz serving as a versatile framework for a range of tasks related to LLM prompting. In your forthcoming revisions, I recommend considering discussing how Parafuzz might be adapted in other areas, perhaps in your future work section.

---

### Official Review · Reviewer_oHTu · 2023-07-06

**Soundness:** 3 good
**Presentation:** 3 good
**Contribution:** 3 good
**Rating:** 5
**Confidence:** 3

**Summary:**

This paper proposes ParaFuzz for discovering if a piece of input text for a poisoned NLP model has backdoor trigger. The idea is is that a paraphrased clean input maintains the same semantic meaning thus the same NLP decision, whereas a poisoned input after paraphrasing will result in a different NLP decision since that the backdoor trigger is lost during the paraphrasing of poisoned texts. ChatGPT is used as the paraphraser. The main contribution of this work is a fuzzing method for automatically discovering the prompts for ChatGPT to paraphrase the input texts to the poisoned NLP model.

**Strengths:**

This work is interesting. Fuzzing for automatically generating prompts to ChatGPT is innovative. As more and more methods rely on ChatGPT, they also require proper designed prompts. It makes sense to automatically generate prompts for ChatGPT and this could be a significant thing.

ParaFuzz outperforms several baselines in the experiments.


**Weaknesses:**

The threat model of ParaFuzz is not always clear and it is possibly weak. As in Fig. 3, ParaFuzz assumes that the NLP model is poisoned, there is clean validation data and there is a crafted poisoned data. The poisoned data were generated by using the backdoor trigger reverse-engineered via PICCOLO on that poisoned model. Briefly speaking, the model has been known to be poisoned and there is even a reversed backdoor available. As a result, what is the contribution the proposed ParaFuzz? In other words, has the difficult part already been resolved before applying the ParaFuzz fuzzing?

There are some parameters (such as threshold and maximum runtime) in ParaFuzz and their configurations are not clear for the fuzzing in experiments.

Fuzzing is often a time-consuming procedure. The runtime of ParaFuzz fuzzing is not reported. Many useful details of the fuzzing process are not mentioned in the experiments.


**Questions:**

1. What’s the relation between ParaFuzz and PICCOLO? Why not compare ParaFuzz with PICCOLO?
2. If there is already a known poisoned set and even a known trigger, what is the contribution of ParaFuzz?
3. How to configure ParaFuzz for prompts fuzzing for experiments in the paper and in general? Are they all the same or different configurations are needed for different NLP model and/or different input text lengths?
4. How much time does the fuzzing take in the experiments? How many iterations are there? How does the coverage change with increased fuzzing iterations?
5. What are the prompts found by the ParaFuzz?


**Limitations:**

Limitations were barely discussed in the paper. The proposed parafuzz has some limitations. ParaFuzz needs to know if the NLP model has changed its decision after the original text has been mutated. While for some NLP tasks it might be easy to check if the NLP decision changed, this is not always trivial. This limits the applicability of ParaFuzz.

---

> ### Author Rebuttal · Authors · 2023-08-10
>
> ### Response to reviewer oHTu
>
> 1. Relation between PICCOLO and PARAFUZZ’s contribution
>
> Please refer to global response section 7.
>
> 2. Configuration of PARAFUZZ
>
> The configuration of PARAFUZZ is the same for all NLP models, even with different input lengths.
>
> 3. Running time and iterations
>
> **Iteration and time:** In experiments we set the maximum iterations to be 300 and the fuzzing process takes 143.88 minutes on average.
>
> **Coverage:** Take style backdoor attack as an example, Figure 10 (see global response page 5) illustrates the variation in coverage with respect to the number of iterations. The validation set contains 200 crafted poisoned sentences. As the number of generated candidates increases during fuzzing, we observe that more poisoned sentences are covered by at least one candidate. Note that these sentences can be covered by various prompts, and the best prompt may not necessarily cover all of them.
>
> 4.  The prompts found by PARAFUZZ
>
> The prompts found by PARAFUZZ are list in Table 4 in appendix.
>
> 5. Limitations
>
> To identify the changes in the model’s predictions, we simply call the model twice: one before the text is paraphrased, and the other after. By comparing the prediction labels from these two calls, we can easily determine if the model’s decision has changed. All ParaFuzz acquires from the model is the prediction labels. This is a minimal requirement compared to the baselines that need to manipulate the embedding layer or access the prediction probabilities.

---

> > ### Comment · Reviewer_oHTu · 2023-08-14
> >
> > As a  test-time poisoned sample detection framework, would 143.88-minute fuzzing time be meaningful? Could you please justify this by also comparing with the baselines.

---

> > > ### Author Response · Authors · 2023-08-15
> > >
> > > The fuzzing process is a pre-test procedure and executed only once. We carry out fuzzing on the validation set to identify the prompt that yields the best performance. Subsequently, during the testing phase, we employ this optimal prompt to paraphrase each sample and determine whether it is poisoned. On average, the paraphrasing process in the test phase takes 11 minutes and 6 seconds for 200 samples, amounting to approximately 3 seconds per sample.
> > >
> > > Please kindly let me know if this addresses your concern.

---

### Author Rebuttal · Authors · 2023-08-10

We sincerely thank all the reviewers for the constructive comments. Please refer to the PDF for details, thanks!

---

### Author Response · Authors · 2023-08-18
**Thank all the reviewers for the time and effort you've dedicated to reviewing our paper**

We would like to express our sincere gratitude to all the reviewers for your help in improving our paper.

We will revise accordingly in the next version, integrating the additional results and discussions in the rebuttal period.

---

### Decision · Program_Chairs · 2023-09-21

**Decision:**

Accept (poster)

**Comment:**

The paper presents ParaFuzz to detect poisoned samples for NLP models. They properly leverage the strong LLMs (e.g., ChatGPT) and demonstrate the efficacy of the proposed method.

The authors removed most of the reviewers' concerns through the successful rebuttal, and finally all reviewers lean to accept the work.